# Nonlinear wavefront shaping with optically induced three-dimensional nonlinear photonic crystals

Shan Liu[1], Krzysztof Switkowski[2,3], Chenglong Xu [4], Jie Tian[4], Bingxia Wang[5], Peixiang Lu [5], Wieslaw Krolikowski[1,3] & Yan Sheng [1]

Generation of coherent light with desirable amplitude and phase profiles throughout the optical spectrum is a key issue in optical technologies. Nonlinear wavefront shaping offers an exceptional way to achieve this goal by converting an incident light beam into the beam (or beams) of different frequency with spatially modulated amplitude and phase. The realization of such frequency conversion and shaping processes critically depends on the matching of phase velocities of interacting waves, for which nonlinear photonic crystals (NPCs) with spatially modulated quadratic nonlinearity have shown great potential. Here, we present the first experimental demonstration of nonlinear wavefront shaping with three-dimensional (3D) NPCs formed by ultrafast-light-induced ferroelectric domain inversion approach. Compared with those previously used low-dimensional structures, 3D NPCs provide all spatial degrees of freedom for the compensation of phase mismatch in nonlinear interactions and thereby constitute an unprecedented system for the generation and control of coherent light at new frequencies.

---

[1] Laser Physics Center, Research School of Physics and Engineering, Australian National University, Canberra, ACT 2601, Australia. [2] Faculty of Physics, Warsaw University of Technology, Warsaw 00-661, Poland. [3] Science Program, Texas A&M University at Qatar, Doha, Qatar. [4] MicroNano Research Facility, RMIT University, Melbourne, Victoria 3001, Australia. [5] School of Physics and Wuhan National Laboratory for Optoelectronics, Huazhong University of Science and Technology, 430074 Wuhan, China. Correspondence and requests for materials should be addressed to Y.S. (email: yan.sheng@anu.edu.au)

Recently, generation of spatially modulated wavefronts via optical frequency conversion, i.e., nonlinear wavefront shaping, has attracted a great deal of research interest[1–7]. Compared with traditional beam shaping processes based on linear optical elements such as spatial light modulators, the added functionality of nonlinear frequency conversion allows manipulation and structuring of coherent light at new frequencies, which opens up exciting new possibilities for important applications including all-optical self-routing[1], volume nonlinear holography[2], generation of entangled states[3], and ultrafast laser technology[8]. Furthermore, the nonlinear wavefront shaping based on alternative approach of structuring the input beam first and then converting it into different frequency, usually does not work due to the difficulty of satisfying the phase-matching requirements with non-Gaussian beams.

Nonlinear optical microstructured materials, such as nonlinear photonic crystals (NPCs) [9–11] and nonlinear metamaterials[4,5], have proved to be excellent candidates for nonlinear wave shaping. In particular, the NPC comprises a nonlinear optical medium with spatially modulated quadratic $\chi^{(2)}$ nonlinearity, which ensures the high efficiency of nonlinear optical interactions via the so-called quasi-phase matching (QPM)[12–14]. Depending on the geometry of interaction, the QPM can be implemented through the schemes of nonlinear Bragg diffraction[15], Čerenkov-type[16] emission and nonlinear Raman-Nath diffractions[17]. In these NPCs the wavefront shaping often relies on transversely patterned nonlinearity, i.e., $\chi^{(2)}$ varies spatially (typically between $+|\chi^{(2)}|$ and $-|\chi^{(2)}|$) in a plane perpendicular to the propagation direction of the incident beam[18–20]. In this respect, the transverse $\chi^{(2)}$ modulation plays a role analogous to that of spatial variation of refractive index in linear optical media. However, while the index pattern imposes just desired amplitude and phase profile on diffracted incident wave, in nonlinear media the spatial $\chi^{(2)}$ pattern leads to frequency conversion with simultaneous amplitude and phase structuring of the emitted new frequencies. Because of analogy with linear diffraction this nonlinear process is called nonlinear diffraction[15,17]. It has been already demonstrated that appropriately designed $\chi^{(2)}$ patterns provide new functionalities including second harmonic (SH) beam-focusing and splitting, as well as the generation of spatially complex optical frequency converted wavefronts such as Bessel, Airy, Semi-Gaussian, and vortex sum frequency beams[1,19–21]. As an example, Fig. 1a illustrates schematically the formation of optical vortices in second harmonic generation (SHG) process. Here the NPC comprises fork-type spatially periodic modulation of the sign of quadratic nonlinearity. When illuminated by a fundamental beam (FF) this nonlinearity grating gives rise to emission of SH waves via nonlinear diffraction. In this particular situation the nonlinear diffraction of a Gaussian FF results in formation of multiple SH beams in form of optical vortices, i.e., donut beams with helical wavefronts. Furthermore, the SHG process fulfills a general conservation law for the orbital angular momentum (OAM)[19], i.e. $l_{SH} = 2l_{FB} + ml_c$, where $l_{SH}$ and $l_{FB}$ represent the OAM of the SH and fundamental beams, respectively ($l_{FB} = 0$ for a Gaussian fundamental beam), $l_c$ is the topological charge of the fork structure, and $m$ is the nonlinear diffraction order of the SH wave.

The most common technique used to fabricate NPCs is the electrical poling of ferroelectric crystals[22]. In this process strong electric field applied to the crystal via patterned electrodes reverses locally direction of spontaneous polarization which amounts to the sign reversal of $\chi^{(2)}$. However, in this traditional approach for the formation of NPCs the nonlinearity modulation is restricted to a single transverse plane. The ability to modulate $\chi^{(2)}$ along the third orthogonal direction is highly sought after and it has been theoretically shown the a 3D modulation would greatly enhance the role of NPCs in many applications including wavefront shaping[23–28]. As displayed in Fig. 1b, a 3D NPC may contain versatile multilayered $\chi^{(2)}$ nonlinearity modulation structures to generate multiple second harmonics with different wavefronts. This is a unique capability that cannot be accomplished with a single low-dimensional nonlinear photonic crystal.

The 3D spatial nonlinearity modulation can be achieved, e.g., by selective amorphisation of crystalline structure of the non-linear medium, with tightly focused femtosecond pulses, as demonstrated recently[29,30]. As the amorphisation decreases the strength of the nonlinearity, one can, in principle, realize arbitrary spatial modulation of $\chi^{(2)}$. However, this process introduces significant scattering losses[30] for interacting beams and hence has limited practical applicability.

We have very recently introduced a breakthrough technique for fabrication of 3D NPCs via ultrafast infrared light induced ferroelectric domain inversion[31]. The technique utilizes nonlinear absorption of light in the optical beam's focal volume. That, in turn, induces a high-temperature gradient and appearance of a thermoelectric field that allows for precise inversion of spontaneous polarization and, consequently, the sign of $\chi^{(2)}$ nonlinearity[32,33]. By carefully controlling the infrared radiation inside a transparent $Ba_{0.77}Ca_{0.23}TiO_3$ ferroelectric crystal, we have realized the first monolithic 3D NPCs and demonstrated their application in nonlinear frequency conversion. As the inversion

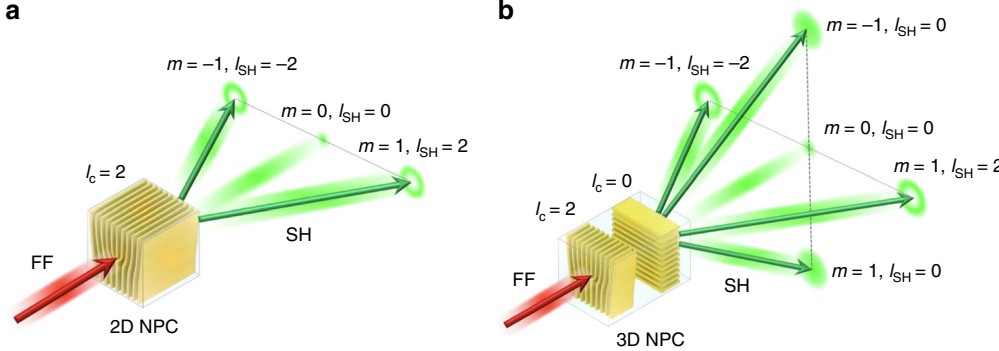

**Fig. 1** Illustrating the concept of nonlinear wavefront shaping in nonlinear photonic crystal (NPC). **a** The 2D NPC contains region of fork-type spatial $\chi^{(2)}$ nonlinearity modulation. When illuminated by a strong Gaussian fundamental beam (FF) the crystal generates second harmonic (SH) wave in form of multiple optical vortices, i.e., donut beam with spiral phase front. The symbols $m$, $l_c$, and $l_{SH}$ represent nonlinear diffraction order, topological charges of the fork structure and the optical vortices, respectively. **b** Being superior to the lower-dimensional structures, a 3D NPC may contain versatile multilayered $\chi^{(2)}$ modulation patterns leading to SH emissions in multiple directions with different wavefronts. Here for simplicity we depict a two-layer structure comprising the fork and one-dimensional grating for generation of donut and Gaussian SH beams

of ferroelectric domains does not induce optical loss, this infrared beam method offers an exceptionally efficient way to fabricate 3D NPCs.

In this letter, we present the first experimental demonstration of novel functionalities of 3D NPCs which are inaccessible in traditional lower-dimensional systems. In particular, we fabricate a number of optically induced nonlinearity patterns in ferroelectric $Ca_{0.28}Ba_{0.72}Nb_2O_6$ (CBN) crystal and employ them for parallel three-dimensional nonlinear wave shaping in quasi-phase matched second harmonic generation. By using transversely overlapping fork, grating, and homocentric ring domain patterns localized at various depths inside a single CBN crystal, we demonstrate the simultaneous conversion of a Gaussian fundamental beam into three pairs of second harmonics with helical, Gaussian, and conical wavefronts, respectively. Furthermore, by making use of longitudinal separation of structures we achieved the first real dynamic control of nonlinear wavefront shaping in 3D NPC. We show that by just shifting longitudinally the focal position of the fundamental beam inside the crystal, we can vary the spatial structure of the emitted SH. Our results clearly show that 3D NPCs represent a major advancement compared to the low-dimensional geometries, opening possibilities for versatile control of nonlinear optical interactions and will contribute significantly to development of structured light sources at new frequencies.

## Results

**Optically induced 3D NPCs.** Direct femtosecond laser writing of 3D ferroelectric domain patterns was accomplished in a z-cut CBN crystal. This is an as grown ferroelectric crystal and as such exhibits multi-domain form featuring 180° domains of submicron size with spatial random distribution[34]. The domain engineering process relies on nonlinear absorption of light in the optical

beam's focal volume, which induces a high-temperature gradient and appearance of a thermoelectric field that forces the spontaneous polarization of the crystal to align along the same direction[31]. The use of infrared femtosecond pulses allowed us to focus the light deep inside the transparent CBN and, consequently, to form various 3D ferroelectric domain structures (See the Methods). In Fig. 2, we depict schematically few different domain patterns and their locations inside the sample. We employed three basic $\chi^{(2)}$ nonlinearity patterns comprising fork grating, regular one-dimensional grating, as well as circular grating. Figure 2a, b depicts the three-layer nonlinear photonic structures, and Fig. 2c shows the two-layer structure of fork and linear grating. In each case, the nonlinearity patterns spatially overlap in the transverse plane but are located at different depths inside the CBN sample. The images in Fig. 2d–f represent 3-D visualization of these actual structures obtained by the nonlinear Čerenkov microscopy[35]. Furthermore, we used the piezoresponse force microscopy (PFM)[36] to confirm that the structures formed in this way are indeed ferroelectric domain patterns and not material modifications such as an optical damage or refractive index change (See Supplementary Note 1 and Supplementary Fig. 1).

In more details, the pattern 1 contains three periodic $\chi^{(2)}$ gratings with charge $l_c = 2$ fork-type dislocation, oriented at 0°, 60°, and 120° with respect to horizontal direction (Fig. 2a, d). The gratings are ~30-μm thick and the distance between neighboring layers is ~40 μm inside the crystal. The second sample of NPC comprises horizontally oriented charge $l_c = 2$ fork pattern, vertically orientated linear grating, and a circular grating (Fig. 2b, e). Finally, the third, shown in Fig. 2c, f, structure consists of two 30-μm-thick layers of nonlinearity modulation in the form of fork-type pattern and linear grating, respectively, mutually rotated by 90°. All these $\chi^{(2)}$ gratings were fabricated to have a periodicity of $\Lambda = 2$ or 3 μm and the

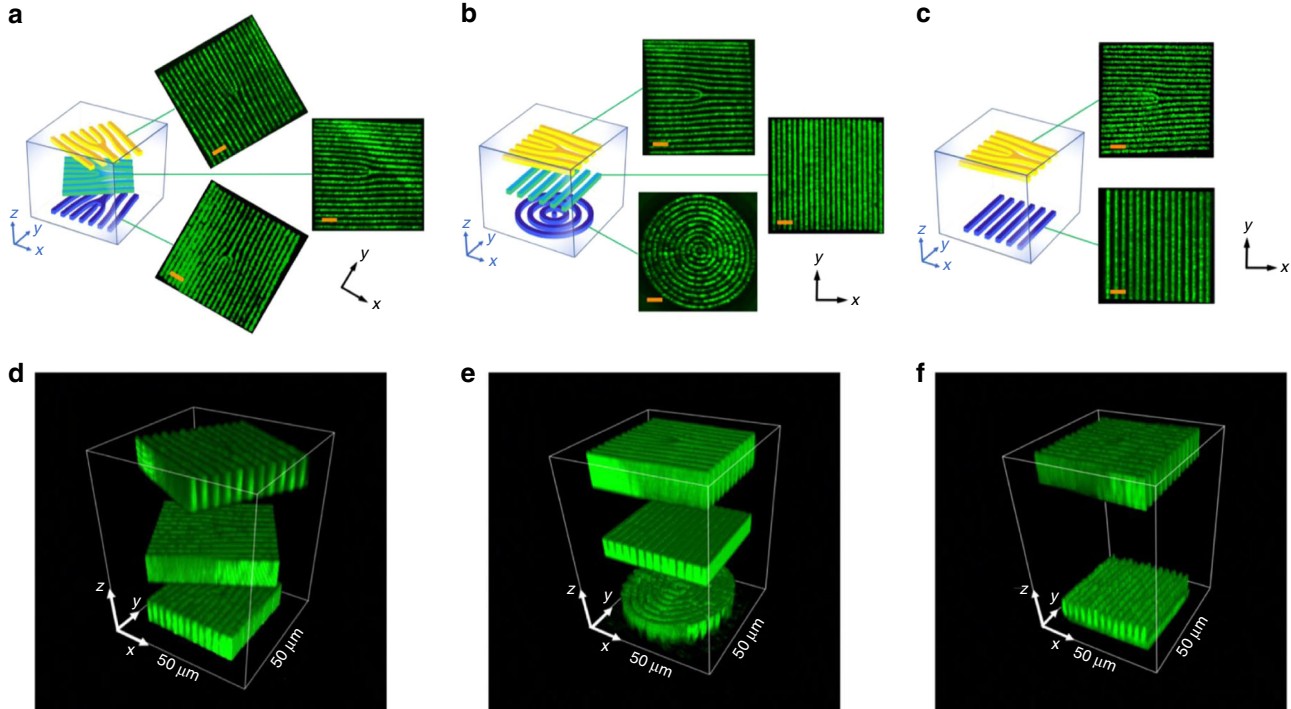

**Fig. 2** Schematic representation of multilayer spatial modulation of $\chi^{(2)}$ nonlinearity in 3D nonlinear photonic crystals. **a** Three-layer fork structures with a rotation of 60° from each other. **b** Fork, linear grating and circular grating. **c** Two-layer fork and linear grating. **d**–**f** The 3-D visualization of these actual structures obtained by the nonlinear Čerenkov microscopy[35]. In these figures the $\chi^{(2)}$ grating period is $\Lambda = 3$ μm, and the topological charge of the fork structure is $l_c = 2$. The three-layer domain structures are located about 23, 69, and 115 μm below the surface of the crystal, respectively in **d** and **e**, and the two layers are located about 23 and 345 μm below the surface in **f**. The length of the scale bar is 10 μm in **a**–**c**

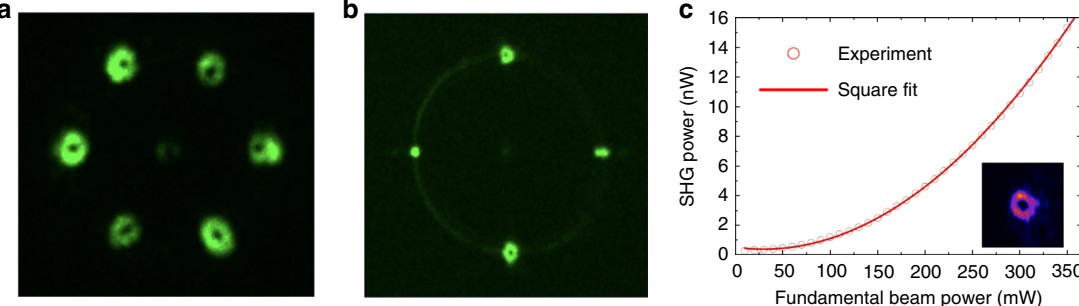

**Fig. 3** Far-field second harmonic patterns emitted from different nonlinear photonic crystals with optically induced and longitudinally (along fundamental beam direction) separated domain patterns. **a** The second harmonic pattern emitted from the three-layer fork-type nonlinear gratings mutually rotated by 60°. **b** The second harmonic from the 3D NPC comprising a fork grating, a one-dimensional linear grating and a circular grating, mutually rotated by 90°. **c** Power of the emitted SH vortex (inset) as a function of the power of fundamental beam. Here the wavelength of the fundamental beam was 1600 nm. The colors of the second harmonic in the figures are false and are used for better visualizations

inverted domain structures were uniform across the whole written area of $60 \times 60 \ \mu m^2$.

**Versatile control of nonlinear interactions with 3D NPCs**. For studies of frequency conversion and wave shaping the 3D NPCs were illuminated by loosely focused fundamental beam propagating along the z-axis, such that the multilayered ferroelectric domain patterns were all simultaneously irradiated. In Fig. 3 we present a few examples of the observed far-field SH patterns. Figure 3a represents SH emission from the three-layered fork structures with dislocation $l_c = 2$ and periodicity $\Lambda = 2 \ \mu m$. One can clearly identify the six vortex beams emitted from the three fork patterns. These vortices represent first-order nonlinear Raman-Nath diffraction patterns. The measured emission angle is 23.9° outside the crystal, agreeing well with the theoretical value of 23.6° (see Method). In fact the higher order nonlinear diffraction patterns are also emitted in the process but they are much weaker compared to the first order. As the fork dislocation is of the second order, the emitted SH vortices carry the topological charge $l_{SH} = \pm 2$. We confirmed the charge by using astigmatic transformation introduced by placing a cylindrical lens in the paths of emitted beams. The resulting intensity distribution (see Supplementary Note 2 and Supplementary Fig. 2) in the focal plane exhibits two parallel dark stripes indicating charge $l_{SH} = \pm 2$.

Figure 3b depicts second harmonic pattern emitted from the three-layered structure comprising fork, linear, and circular gratings of the same periodicities ($\Lambda = 2 \ \mu m$). One can clearly identify two donut beams along the vertical and two localized intensity peaks along horizontal directions, respectively. They all lay on a circle representing conical SH beam emitted by circular domain pattern. The graph in Fig. 3c shows dependence of the power of one of generated vortices, as a function of input power of the fundamental beam. This plot confirms well-known fundamental quadratic relation between intensity of the SH and fundamental beam.

Simultaneous appearance of multiple SH patterns of almost equal intensity generated in the volume of 3D NPC results from the fact that the fundamental beam is not being depleted in the interaction process. Therefore, this beam experiences nonlinear diffraction on each of the embedded nonlinear holograms with almost constant power. Furthermore, since the input beam consists of train of pulses, the SH emission from these two or three holograms (nonlinearity patterns) is completely independent as their separation exceeds the pulse length. This, in turn, gives rise to the total intensity pattern as an incoherent superposition of contributions from each of the layer separately. In fact, this process can be analyzed theoretically by employing

simple model for second harmonic generation in the undepleted pump regime. In this, low conversion efficiency regime, the far-field emitted SH is expressed by the Fourier transform of the nonlinearity modulation[37]. In Fig. 4, we illustrate results of the calculations for the three-layer structure of fork patterns (Fig. 2a, d) and assuming cw Gaussian fundamental beam. We used the actual (a non-ideal) nonlinearity profiles as recorded experimentally. Six charge 2 vortices are visible. Notice the visible tendency of some calculated vortices to split into two fundamental vortices. This is a well-known effect of instability of high order vortices which can easily split into fundamental vortices under small coherent perturbation or distortion of the fork pattern[38,39]. Interestingly, the vortices observed in experiment did not split displaying well-defined dark core. The most likely reason of experimentally observed robustness of charge 2 vortices is that we use short (200 fs) pulses not cw in fundamental beam. The broader spectrum of the pump leads to smearing of the vortices making them effectively insensitive to small scale variations of the nonlinearity pattern.

**Dynamic control of nonlinear optical waves with 3D NPCs**. To better demonstrate the multilayer character of ferroelectric domain pattern in wavefront shaping we optically wrote two planar nonlinearity structures containing mutually orthogonal (as far as the spatial orientation is concerned) fork-type grating and simple linear grating with the same periodicities. The structures were laser-induced inside the CBN sample and were separated in longitudinal direction such that could be independently accessed with tightly focused beam. The focusing position of the fundamental beam was fixed and then the sample was moved along the beam direction (z-axis) such that the light focal spot was translated through the sample. Figure 5 demonstrates emitted second harmonic beams for two sample locations separated by approximately 322 μm inside the crystal. It is clear that either two Gaussian-like or two vortex second harmonic beams are generated depending on the axial position of the sample. However, the images also show some visible cross-talk between two emitted patterns. It is caused by the axial elongation of the focus of the incident beam deep inside the crystal, which is caused by the spherical aberration introduced by the high refractive index of the CBN crystal[40] such that even for nominal focusing of the beam at the first structure, the periphery of the focus still partially overlaps with the second structure generating the spurious (crosstalk) SH signal. To the best of our knowledge this is the first time to obtain dynamic nonlinear wavefront shaping[41] with 3D nonlinear photonic crystals. Thereby, this work represents a significant

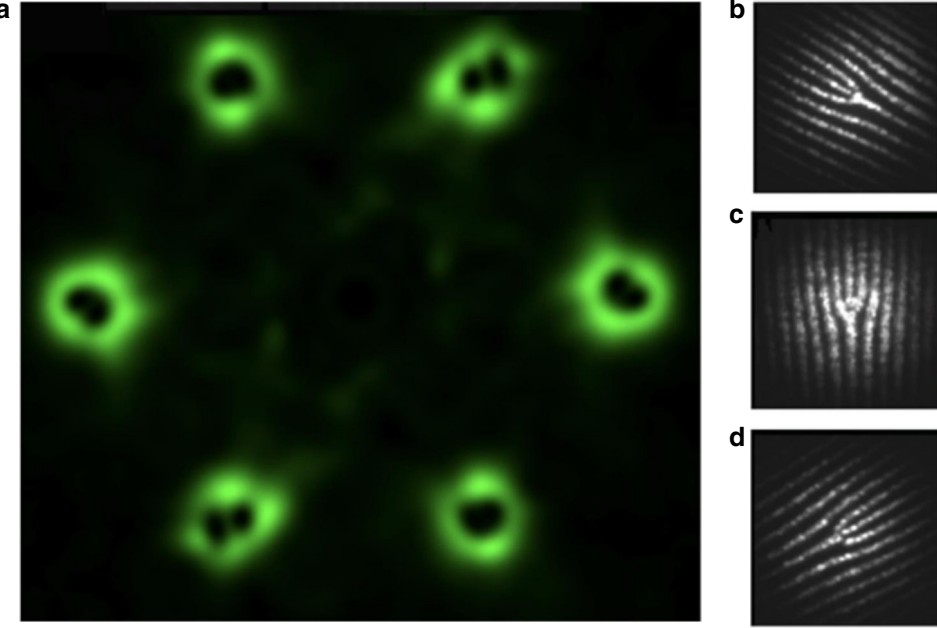

**Fig. 4** Theoretical modeling of the second harmonic generation from 3D nonlinear photonic crystal. **a** Calculated far-field second harmonic intensity distribution emitted from the three-layer fork nonlinearity pattern illuminated by Gaussian fundamental beam at 1000 nm. **b–d** The $\chi^{(2)}$ patterns used in the calculations

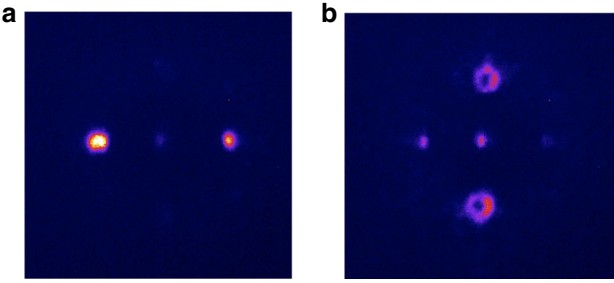

**Fig. 5** Dynamic control over the second harmonic waves in a single 3D nonlinear photonic crystal. This is achieved by selectively exciting one-dimensional nonlinearity grating (**a**) and fork-type nonlinearity grating (**b**), that are located at different depths in the crystal and with 90° mutual azimuthal orientation. The separation distance between both patterns was approximately 322 μm

progress that may open new possibilities in many fields such as laser display, optical trapping, and imaging.

## Discussion

We fabricated 3D nonlinear photonic crystals by using our femtosecond-laser domain inversion approach in $Ca_{0.28}Ba_{0.72}Nb_2O_6$ ferroelectrics, and have demonstrated unprecedented control over the nonlinear wave emissions from these crystals thanks to the full flexibility of 3D nonlinearity modulation. Using a three-layer nonlinear photonic structure comprising fork grating, regular 1D grating, as well as circular grating embedded inside the crystal, we demonstrated a simultaneous conversion of a fundamental Gaussian beam into second harmonic vortex, Gaussian, and conical beams. In addition, thanks to the longitudinal separation between layers of domain patterns, we demonstrated the first dynamic control over the nonlinear interactions in 3D nonlinear photonic crystals. In fact, as ferroelectric domains can be imprinted independently to form various patterns at different depths, the 3D nonlinear photonic crystals

pave the way for manipulating target-specific features of nonlinear interactions without affecting the others. Thus, the results reported here constitute an important contribution towards development of advanced coherent light sources at new spectral range, which are at the core of many optics-related fundamental research and industrial applications. In order to exploit further all advantages of 3D character of nonlinearity modulation enabling quasi-phase matched interaction in arbitrary direction, our future efforts will be directed towards realization of complex nonlinearity structures with continuous spatial modulation of the nonlinearity along propagation direction, such as the previously proposed twisted nonlinear photonic crystals[26,28]. Structures of such complexity are beyond reach of traditional techniques and can only be fabricated by employing our optical domain reversal method.

## Methods

**Optical writing of 3D ferroelectric domain patterns.** We used a z-cut as-grown sample of calcium barium niobate crystal. As the crystal was unpoled it exhibited a random spatial distribution of antiparallel ferroelectric domains. The multi-domain character was confirmed by strong conical (Čerenkov) emission of second harmonic signal upon the propagation of fundamental beam propagating along the crystal z-axis. Based on our earlier works with the samples from the same crystal boule, the random domains are mostly on the submicron scale[34].

The infrared writing optical beam originated from a femtosecond oscillator (MIRA, Coherent) operating at 800 nm (short enough to excite the two-photon process), with a pulse duration of 180 fs and a repetition rate of 76 MHz. The optical beam was focused by a 50× microscope objective (NA = 0.65) into the sample with an estimated beam diameter of ~1 μm. The illumination of tightly focused femtosecond laser leads to the alignment of the originally random domains along the same direction within the focus area. To this end the CBN sample was mounted on a translational stage that could be positioned in three orthogonal directions with a resolution of 100 nm. The power of the incoming laser beam could be continuously varied by using a half wave plate followed by a polarizer.

The multilayer ferroelectric domain patterns were written layer by layer and from bottom to top. For domain inversion in each layer, the sample was translated against the beam propagation direction so that the position of the focal region of the writing beam moved at an average speed of 10 μm/s over a distance of 20 μm. After that a shutter blocked the beam and the sample was transversely translated to the next location where the process was repeated. In order to produce stripe-like areas of inverted domain, the distance between the neighboring domain spots was kept small (~0.5 or 1 μm) causing them to merge. The average power of the writing

beam was selected to be slightly lower than the optical damage threshold, and was 505 mW, 370 mW, and 280 mW for the three layers located at $z = 115, 69$, and 23 μm below the surface, respectively. The influence of laser power on the resulting domain patterns was discussed in our earlier work[31].

The typical sizes of the laser-written domain area are $60 \times 60$ μm$^2$ in the transverse plane (XY in our case). The length of inverted domains in each layer is ~30 μm, measured by using the Čerenkov second harmonic microscopy. The maximal writing depths that could be reached from one side of the sample was ~150 μm, and was restricted mainly by the maximal laser power available in the experiment. Furthermore, the focusing condition of the femtosecond laser beam gets worse with the depth due to aberrations, which also makes it difficult to write deep inside materials with high refractive indices. In order to create domain layers that are sufficiently far from each other for dynamic nonlinear wave shaping, we wrote a layer of domains from one side of the sample, and then wrote the other layer from the other side. To simplify localization of nonlinearity structures and their alignment in our experiment, rectangular frames made of optically damaged spots were created surrounding the laser-written ferroelectric domain patterns. The frames also served as alignment markers to facilitate precise spatial overlap of the two or three layers in the transverse plane.

**Nonlinear wavefront shaping experiment**. The incident laser beam was derived from the Coherent Chameleon + Chameleon OPO system delivering 200 fs pulses at 80 MHz repetition rate with the wavelength being tunable between 800 and 1600 nm. The beam propagated along the z-axis of the sample and was linearly polarized to generate the second harmonic via nonlinear coefficient $d_{31}$. The beam was loosely focused into the NPCs and the beam size was slightly exceeding 70 μm in diameter, to ensure the illumination of the whole area of nonlinearity modulation ($60 \times 60$ μm$^2$). A short-pass filter was used to block the fundamental. The emitted second harmonic signal was projected on to a screen placed behind the sample and then recorded by a CCD camera. The SH images shown in Fig. 3 are recorded at a fundamental wavelength of 1600 nm and with an average power of 350 mW. We confirmed the used laser illumination in SHG experiment did not alter or produce additional ferroelectric domain patterns. The different divergence properties of the SH vortex beams in Fig. 3a, b are caused by different focusing condition of the fundamental Gaussian beam (with numeric aperture NA = 0.3 to obtain image in Fig. 3a and NA = 0.1 for Fig. 3b). As to the SHG shown in Fig. 5, the fundamental beam at 1550 nm was focused with a microscope objective of numerical aperture NA = 0.25.

**Emission angle of the second harmonic waves**. The periodic nonlinear structures lead to second harmonic generation via the nonlinear Raman-Nath diffraction, the emission angle of which is determined by the transverse phase matching condition, i.e., $k_2 \sin\theta = mG$, where $k_2$ is the wave vector of the SH beam, $\theta$ is internal emission angle, $m$ is an integer denoting the order of nonlinear diffraction, and $G$ is reciprocal lattice vector corresponding to the $\chi^{(2)}$ modulation pattern. Taking Snell's law into account, we obtain the external emission angle of the first-order nonlinear Raman-Nath diffraction $\alpha = \sin^{-1}(m\lambda_2/\Lambda)$, where $\lambda_2$ is the wavelength of the second harmonic and $\Lambda$ is the period of $\chi^{(2)}$ modulation outside the crystal. With the used parameters of $\lambda_2 = 800$ nm, $m = 1$, and $\Lambda = 2$ μm, we obtain a theoretical external emission angle of $\alpha = 23.6°$.

The efficiency of transversely matched SHG is rather weak. Only the full-phase matching ensures monotonic flow of power from fundamental to second harmonic beams. Such condition can be relatively easily achieved in our multilayer domain structures by choosing the wavelength of fundamental beam to make Čerenkov second harmonic emission angle ($\alpha_{\check{C}} = \cos^{-1}(n(\lambda_2)/n(\lambda_1))$)[16,17], coincides with one of the Raman-Nath nonlinear diffraction orders. Here $n(\lambda_1)$ and $n(\lambda_2)$ are refractive indices of the fundamental and SH waves, respectively. In this nonlinear Bragg diffraction regime, Čerenkov process ensures fulfillment of the longitudinal, while nonlinear Raman-Nath diffraction transverse phase matching, giving rise to monotonic growth of the intensity of SH with propagation distance.

## Data availability
The data that supports the results within this paper and other findings of the study are available from the corresponding authors upon reasonable request.

## Code availability
The custom code and mathematical algorithm used to obtain the results within this paper are available from the corresponding authors upon reasonable request.

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

## Acknowledgements

This work was supported by the Qatar National Research Program (Grants # NPRP8-246-1 060 and NPRP9-020-006), the Australian Research Council, and Foundation for Polish Science co-financed by the European Union (Grant HOMING POIR 04.04.00-00-5E4E/18-00).

## Author contributions

Y.S. conceived the project; S.L. designed and fabricated structures; K.S. conducted frequency doubling experiments; C.X. performed piezoresponse force microscopy measurements; Y.S. and W.K. coordinated the project. All authors made significant contributions to analysis of data, discussions, and writing the manuscript.

## Additional information

**Competing interests:** The authors declare no competing interests.

