## [Peer Review File · Nature Communications]

Reviewers' Comments:

Reviewer #1:

Remarks to the Author:

This is an interesting and original paper, in which the authors utilize a recent capability they developed, of 3D modulation of the second order nonlinear coefficient, for achieving nonlinear optical wavefront shaping.

Up until recently, nonlinear photonic crystals were limited to only two-dimensional modulation, but last year this group of authors, and another group from Nanjing University, reported independently a breakthrough capability of modulating the nonlinearity in three dimensions, by relying on femtosecond laser to either locally pole (ref 29) or erase (ref 27) the nonlinear coefficient. In this paper, the authors take this new capability one step further and use the 3D modulation for nonlinear wavefront shaping. They fabricated and measured the nonlinear response of a series of crystals that include different structures for shaping the nonlinearly generated waves, at different locations along the propagation axis of the crystal. In this manner, they generated at the second harmonic vortex beams, conical beams, Gaussian beams travelling in different directions, and more. They also show dynamic capability, by focusing the pump into a different locations in the crystal, that generate different corresponding shapes at the second harmonic.

As far as I know, this is the first experimental report on nonlinear beam shaping with 3D modulated nonlinear photonic crystals. Since this demonstration opens very interesting new possibilities in nonlinear optics and in wavefront shaping, I recommend accepting the paper for publication in Nature Communications, pending revisions as detailed below:

The structures that were fabricated are in fact a combination of separate two-dimensional structures, stacked inside the crystal along the Z axis. In each one of these separate structures, there is no dependence of the modulation pattern on the propagation coordinate. I believe that this design does not fully utilize the capability of modulating in the third dimension. Specifically, the current design is not phase matched in the propagation axis, and therefore exhibits low conversion efficiency (nano-Watts levels of second harmonic for hundreds of milli-Watts of the pump, as shown in Fig. 3c) into the shaped wavefront. I wonder if the authors can discuss or demonstrate more efficient schemes, based nonlinear volume holography, that require modulation along the Z axis? Can the authors refer to these possibilities and compare them to the designs that they chose?

Fig 3c – what was the pump beam size and what was the peak power? Also, for comparison, what would be the phase matched conversion efficiency in a crystal with similar length and pump beam.

The authors write "To the best of our knowledge this is the first time to obtain dynamic nonlinear wavefront shaping with nonlinear photonic crystals in experiment." – Perhaps this is correct for 3D crystals, but for 2D crystals it is not correct. Dynamic shaping was already shown in the past in nonlinear photonic crystals and should be mentioned, see Trajtenberg-Mills et al, Optica 4, 153 (2017)

Reviewer #2:

Remarks to the Author:

The authors demonstrate experimentally for the first time nonlinear beam shaping by quadratic nonlinear crystals with 3D engineered domain structures. They use a femtosecond laser writing technique, that was recently developed by them, to create the 3D structures in the materials. Specifically, experimental demonstrations include nonlinear emission from three layer or two layer structures. They show how different beams can be created simultaneously and how the depth difference of the different layers can be used to achieve dynamic control over the generated

beams.

I find that the paper is well written. It reports for the first time on a very important application of recently developed 3D laser writing technique of quadratic domain structures. I am certain that this demonstration will have a very strong impact on nonlinear optics community and on various laser applications. Therefore, I find it highly suitable for publication in Nature Communications Journal. I have only a few suggestions listed below for minor corrections that might improve the manuscript.

Suggestions for minor corrections:

1. Lines 62-63: "Furthermore, the m -th nonlinear diffracted order by a fork structure of the topological charge l_c is a vortex beam with a "charge" $l_{SH} = ml_c$, representing $2n l_{SH}$ azimuthal phase modulation."

I think this should be explained better also for non experts, as it is a key concept for understanding the results.

2. The sub-figures in fig 2(a)-(c) would better be enlarged so it would allow to see the details better.

3. Emphasize the difference in parameters for laser writing and for SHG experiments.

4. The specific demonstrations that were shown besides the "dynamic" one can be achieved also with 2D modulations of the three structures. add some more insight on benefit of the 3 layer approach or better yet its extension to true volume nonlinear shaping.

Reviewer #3:

Remarks to the Author:

In the paper entitled " Nonlinear wavefront shaping with optically-induced three-dimensional nonlinear photonic crystals" 1, Shan Liu et al demonstrate second harmonic generation in nonlinear photonic structures fabricated by using the femtosecond laser domain inversion technique, which has been developed by the same team. They perform frequency doubling in nonlinear grating comprising of fork, circular and one-dimensional nonlinear gratings fabricated in the same CBN crystal. The experiments demonstrated that one can change the nonlinear diffraction pattern by shifting the focal point of the fundamental beam along the propagation direction.

The presented results are interesting and I think the paper may deserve publishing in the Nature communication. However, my feeling is that prior to publication authors should address the following issues:

1. The paper essentially based on the Ref 29, in which authors have already reported the fabrication of the 3D NPCs via the femtosecond-laser domain inversion. That is the fabrication several NPCs in the same crystal reported in the current paper looks like an incremental improvement of the results obtained in Ref 29. Authors should clearly explain what is new in the current paper.

2. It is unclear from the text what was the conversion efficiency. Since the gratings are essentially transverse with respect to the fundamental beam, one may expect that the interaction length does not exceed several microns. It is definitely enough to register the SHG signal, however application potential of such a nonlinear optical device may be questionable.

Response to Reviewer #1

As far as I know, this is the first experimental report on nonlinear beam shaping with 3D modulated nonlinear photonic crystals. Since this demonstration opens very interesting new possibilities in nonlinear optics and in wavefront shaping, I recommend accepting the paper for publication in Nature Communications, pending revisions as detailed below:

Reviewer:

The structures that were fabricated are in fact a combination of separate two-dimensional structures, stacked inside the crystal along the Z axis. In each one of these separate structures, there is no dependence of the modulation pattern on the propagation coordinate. I believe that this design does not fully utilize the capability of modulating in the third dimension. Specifically, the current design is not phase matched in the propagation axis, and therefore exhibits low conversion efficiency (nano-Watts levels of second harmonic for hundreds of milli-Watts of the pump, as shown in Fig. 3c) into the shaped wavefront. I wonder if the authors can discuss or demonstrate more efficient schemes, based nonlinear volume holography, that require modulation along the Z axis? Can the authors refer to these possibilities and compare them to the designs that they chose?

Authors' Response

We appreciate the referee's insightful comments. What we present in this manuscript is the first demonstration of superiorities of 3D nonlinear photonic crystal (NPC) in nonlinear wavefront shaping and the proposed concept is quite straightforward. Stacking a number of 2D nonlinearity modulation patterns in the same crystal, along the propagation direction of fundamental beam (Z axis in our case), leads to emission of second harmonic waves having their wavefronts spatially shaped in arbitrary forms (e.g., vortices, Gaussian and conical beams in our experiment). While in this simple design the individual domain stacks play their roles independently, our 3D NPC still represents a significant advancement in nonlinear wavefront shaping showing new effects such as the dynamical wave shaping by tuning the focus position along the beam propagation direction, that cannot be realized in low dimensional structures. We agree with the referee that going over to more complex 3D structures will open up possibilities to observe entirely new effects provided by nonlinear holograms. The experimental realization of such spatially complex 3D structures is a subject of our ongoing efforts. However, it requires significant improvement and optimization of our femtosecond laser ferroelectric domain writing technique, which we hope to achieve in the near future. Hence, in this work we focused on demonstrating the application of simpler multilayer 3D structures. We commented on the formation of extended 3D nonlinearity structures, including those representing true nonlinear holograms, in the concluding parts of the revised version of the manuscript (Page 14, highlighted in yellow)

Although the current design does not rely on a periodic nonlinearity modulation along the beam propagation direction (Z axis), the observed second harmonic generation processes can still be fully-phase matched at certain wavelengths, at which the angle of Čerenkov SH emission overlaps with the Raman-Nath nonlinear diffraction. In the fully phase matched

regime higher conversion efficiency will be achievable by fabricating high quality, thicker domain structures. For example, with nonlinear gratings having period of $2\ \mu\text{m}$, the SHG is fully phase matched via 1st order nonlinear diffraction at the fundamental wavelength of $1.76\ \mu\text{m}$. Since the longest wavelength available from our laser system was $1.6\ \mu\text{m}$, the results depicted in Fig.3 (c) were obtained at this particular wavelength. The relevant discussion is provided in the Method section (Emission angle of the second harmonic waves, highlighted in yellow, pages 17 and 18).

Reviewer:

Fig 3c – what was the pump beam size and what was the peak power? Also, for comparison, what would be the phase matched conversion efficiency in a crystal with similar length and pump beam.

Authors' Response

The pump beam size was slightly exceeding $70\ \mu\text{m}$ in diameter, to ensure the illumination of the whole area of nonlinearity modulation (60×60 microns) and the used peak power was just below $22\ \text{kW}$. We add this information in the Method section (Nonlinear wavefront shaping experiment, page 16).

As far as the conversion efficiency in Fig.3c is concerned, we estimated it (using experimental data for longitudinal phase mismatch $\Delta k=0.12\ \mu\text{m}^{-1}$, and Eq. (6) in Ref [16]) to be roughly 3 times lower than that in the fully phase-matched case. This means that by going over to nonlinear Bragg regime and making structure longer one can substantially increase conversion efficiency.

Reviewer:

The authors write "To the best of our knowledge this is the first time to obtain dynamic nonlinear wavefront shaping with nonlinear photonic crystals in experiment." – Perhaps this is correct for 3D crystals, but for 2D crystals it is not correct. Dynamic shaping was already shown in the past in nonlinear photonic crystals and should be mentioned, see Trajtenberg-Mills et al, Optica 4, 153 (2017)

Authors' Response

We rephrased the sentence into "To the best of our knowledge, this is the first time to obtain dynamic nonlinear wavefront shaping with 3D nonlinear photonic crystals". We also cited the work mentioned by the referee as our Ref. 41.

Response to Reviewer #2

We thank the referee for a very positive evaluation of our work. Below we address the reviewer's comments.

Reviewer

1. Lines 62-63: "Furthermore, the m-th nonlinear diffracted order by a fork structure of the

topological charge l_c is a vortex beam with a "charge" $l_{SH}=ml_c$, representing $2\pi l_{SH}$ azimuthal phase modulation." I think this should be explained better also for non experts, as it is a key concept for understanding the results.

Authors' Response:

We rewrote the relevant part of the text providing more details. The new sentence reads now: "Furthermore, the SHG process fulfils a general conservation law for the orbital angular momentum (OAM)¹⁹, i.e. $l_{SH}=2l_{FB}+ml_c$, where l_{SH} and l_{FB} represent the OAM of the SH and fundamental beams, respectively ($l_{FB}=0$ for a Gaussian fundamental beam), l_c is the topological charge of the fork structure, and m is the nonlinear diffraction order of the SH wave." (Page 3, highlighted in yellow)

Reviewer

2. The sub-figures in fig 2(a)-(c) would better be enlarged so it would allow to see the details better.

Authors' Response:

We have enlarged Fig 2 (a)-(c) in the revised manuscript.

Reviewer

3. Emphasize the difference in parameters for laser writing and for SHG experiments.

Authors' Response:

The laser power densities were much higher for laser writing of ferroelectric domain patterns than those used for SHG experiment. Typically, for laser writing the laser beam was tightly focused (with $NA=0.65$ objective lens in this work). The average power of the writing beam was selected to be slightly lower than the optical damage threshold. Specifically, it was 505 mW, 370 mW, and 280 mW for the layers located at $z=100$, 60, and 20 μm below the sample surface, respectively. For SHG experiment, the incident laser beam was loosely focused (e.g. with $NA=0.1$, 0.3 objective lens or lens with a focal length of 50 or 75 mm). The highest laser power used for SHG was about 350 mW in this work. We confirmed that the laser used in the SHG experiment did not alter or produce additional ferroelectric domain patterns. We emphasized the difference in the revised manuscript (Method, Nonlinear wavefront shaping experiment, highlighted in yellow on page 17).

Reviewer

4. The specific demonstrations that were shown besides the "dynamic" one can be achieved also with 2D modulations of the three structures. Add some more insight on benefit of the 3 layer approach or better yet its extension to true volume nonlinear shaping.

Authors' Response:

We thank the referee for this comment. Because of experimental limitations (optical power of the writing beam and positioning accuracy) in this work we restricted ourselves to relatively simple multilayer structures. However, it is worth stressing that even the stacking of multiple domain patterns in a single crystal cannot be realized with a standard electrical

poling, which can only result in a single layer of nonlinearity pattern originating at the surface of the sample. We are working on upgrading our laser-writing system to reach in the future the fabrication of sophisticated 3-D structures, with spatial nonlinearity modulation varying continuously in all 3 dimensions, such as the true three-dimensional nonlinear holograms. In the revised manuscript we expanded the introductory as well as concluding paragraphs (Pages 6 and 14, highlighted in yellow), to emphasize the benefits of multilayer nonlinear structures and discuss the extension of our work to true 3D nonlinear holograms with continuing modulation of nonlinearity along the propagation direction.

Response to Reviewer #3

We thank the referee for finding our results interesting and deserving publication in Nature Communications. We address below the referee's concerns. We hope that our response and revision of the manuscript will convince the referee that our work does constitute an important step forward from our previous work and promises the realization of new functional devices in nonlinear photonics.

Reviewer:

1. The paper essentially based on the Ref 29, in which authors have already reported the fabrication of the 3D NPCs via the femtosecond-laser domain inversion. That is the fabrication several NPCs in the same crystal reported in the current paper looks like an incremental improvement of the results obtained in Ref 29. Authors should clearly explain what is new in the current paper.

Authors' Response:

Our earlier work (Ref [31] in the revised manuscript) concentrated on the first fabrication of 3D nonlinear photonic crystal via our own ultrashort laser pulse-induced ferroelectric domain reversal. In that paper, we focused on the novel and versatile optical method to create spatial modulation of nonlinearity in three dimensions. In the current manuscript, we just have employed our domain inversion technique to fabricate simple but unique structures (which cannot be replicated by any other techniques) to demonstrate the novel functionalities of the 3D nonlinearity pattern. As such, our work provides the experimental proof of principle of a parallel 3-dimensional nonlinear wave shaping via quasi-phase matched frequency conversion. We revised the introductory part of the paper to emphasize the novelty of this work (Page 6, highlighted in yellow). Similarly, in the revised concluding paragraph of the paper we discussed the significance of three-dimensional nonlinear structures, in particular, those with sophisticated spatial modulation of nonlinearity representing the true nonlinear holograms (Page14, highlighted in yellow).

Reviewer

It is unclear from the text what was the conversion efficiency. Since the gratings are essentially transverse with respect to the fundamental beam, one may expect that the interaction length does not exceed several microns. It is definitely enough to register the

SHG signal, however application potential of such a nonlinear optical device may be questionable.

Authors' Response:

We are grateful for this review's comment. The conversion efficiency was not high in this work. The reason is, as the review noted, the very short interaction distance (roughly about 30 μm in this work). Here, in order to create more layers of ferroelectric domain structures inside the finite depth of the crystal, we did not produce very long domains. The domain lengths are also restricted by the focusing condition of the femtosecond laser beam, which gets worse with the depth due to spherical aberrations. This is a common issue in femtosecond laser writing, especially for materials with high refractive indices and writing deep below the surface of the medium. However, this adverse effects can be minimized by taking special measures such as pre-shaping the beam with spatial light modulator and using the selected polarization to avoid focus splitting such that much longer samples can be fabricated for the sake of higher conversion efficiencies. We will explore these methods in our future work.

We would like to point out that while the nonlinear gratings are transverse, it is still possible to obtain fully phase matched second harmonic generation at certain wavelength, where the angle of Čerenkov second harmonic emission (along which the longitudinal phase matching condition is satisfied) overlaps with the transverse Raman-Nath nonlinear diffractions. For example, with nonlinear gratings having period of 2 μm , the SHG is fully phase matched via 1st order nonlinear diffraction at the fundamental wavelength of 1.76 μm . We provide the relevant discussion on this point in the Method section (Emission angle of the second harmonic waves, highlighted in yellow, pages 17 and 18).

Reviewers' Comments:

Reviewer #1:

Remarks to the Author:

I appreciate the authors efforts to revise the paper according to my comments and suggestions. Most of the issues were answered in a satisfactory manner.

I wish that the authors could implement their method to fully utilize the three available dimensions for nonlinear beam shaping, rather than stacking two-dimensional patterns along the Z direction. However, I understand that this requires further optimization of their process, beyond what is available now.

I therefore don't have any further comments, and I recommend accepting the paper to Nature Communications.

Reviewer #2:

Remarks to the Author:

I find that the authors revised the manuscript appropriately. Well done for the good achievement!

Reviewer #3:

Remarks to the Author:

Authors have properly addressed all issues raised by Reviewers. I recommend this paper for publication.